# Radiological and Periodontal Evaluation of Stock and Custom CAD/CAM Implant Abutments—A One-Year Follow-Up Study

Ivica Pelivan [1] , Ivan Šeparović [2], Marko Vuletić [3,4,]*, Nikša Dulčić [1] and Dragana Gabrić [3,4]

1 Department of Removable Prosthodontics, School of Dental Medicine, University of Zagreb, 10000 Zagreb, Croatia
2 Digital Smile Academy Clinic, 10000 Zagreb, Croatia
3 Department of Oral Surgery, School of Dental Medicine, University of Zagreb, 10000 Zagreb, Croatia
4 Department of Oral Surgery, University Dental Clinic, University Hospital Center Zagreb, 10000 Zagreb, Croatia
* Correspondence: mvuletic@sfzg.hr

**Abstract:** Implant abutment selection is an important step in implant treatment to restore one or more lost teeth. The aim of this study was to compare stock and individual CAD/CAM full-form abutments after one year in function. A total of 64 subjects with one missing tooth were divided into two groups according to the type of abutment: 34 patients were given a stock abutment, and 30 an individual CAD/CAM abutment. Patients were scheduled for check-ups seven days after functional loading and after four, eight, and twelve months. Peri-implant soft tissue status was checked at every check-up by monitoring parameters traditionally used in similar studies: plaque index; bleeding on probing; and probing depth. To assess the stability of the bone tissue, radiological methods of measuring the amount of bone level compared to the implant shoulder were used. When needed, data were analysed by $\chi^2$ test or by Fisher's exact test. The normality of the distribution of quantitative measurements (properties) was tested by the Shapiro–Wilk test. Differences in the distribution of quantitative variables frequencies were analysed by Student's *t*-test. Student's *t*-test was used for repeated measurements, Mann–Whitney's U test and ANOVA test for repeated measurements, and Friedmann's two-way analysis of variance for repeated measurements. The predictive values of the chosen variables on the ABI index were assessed by the logistic regression model (Enter method). The results of this study showed that the impact of the abutment type (individual CAD/CAM or stock) on the average bleeding on probing was significant, especially after eight or twelve months. However, the abutment type did not show a significant correlation with the total crestal bone loss. The level of oral hygiene showed a significant correlation with the average bleeding on probing. The influence of smoking cigarettes on the total crestal bone loss evaluation was also significant. Overall, from a clinical perspective, custom CAD/CAM abutment performed slightly better than stock abutments during the one-year follow-up.

**Keywords:** dentistry; implantology; dental implants; single-tooth dental implant; dental implant-abutment design; computer-aided design; computer-aided manufacturing; oral hygiene indexes; periodontal indexes

## 1. Introduction

Implant abutment selection is an important step in implant treatment of toothlessness, i.e., in the restoration of one or more lost teeth. Abutments can generally be divided into two main groups: stock and custom [1]. For several years, stock abutments, which implant producers offer in various forms, sizes, and angles of inclination, have been the only option available for clinicians.

The advantages of using CAD/CAM abutments are many and exceed the advantages of using stock abutments. CAD/CAM abutments allow the clinician to individualise the abutment parameters in implant rehabilitation, respecting the soft tissue and maintaining excellent mechanical characteristics. However, some advantages associated with the use of

stock abutments remain, such as the risk of corrosion, time spent, cost, and in vitro assessed fit of the implant [2]. Disadvantages of stock abutments include limited adaptation possibility, unsatisfactory emergence profile, the predetermined position of abutment–prosthetic replacement connection, cylindrical non-anatomic form, and a lower possibility of loading [3]. They result in unsatisfactory contours of prosthetic replacement and insufficient soft tissue support [4]. Many researchers confirmed that the use of stock abutments for cement-retained crowns is no longer justified because residual cement cannot be removed efficiently [5].

Custom abutments that are fabricated individually adapt to soft tissues and respect the biomechanical properties of individual patients [6]. Such an abutment provides adequate support for soft tissues, and the connection between the abutment and a cement-retained crown is moved more towards coronal, which makes residual cement removal easier during the cementation of a prosthetic restoration [7]. Custom abutments can be fabricated in two ways: by means of wax modelling and casting technique and by means of CAD/CAM technique in full anatomical form. The main design differences between stock abutment and CAD/CAM custom abutment are shown in Figure 1. The list of abbreviations used in the text is shown in Table 1.

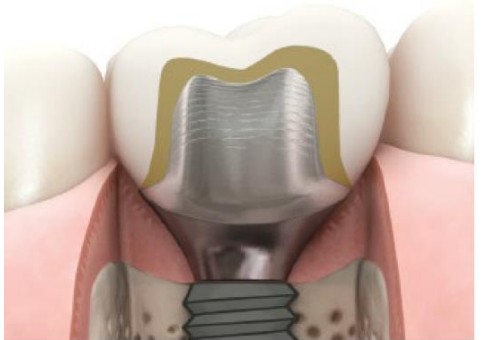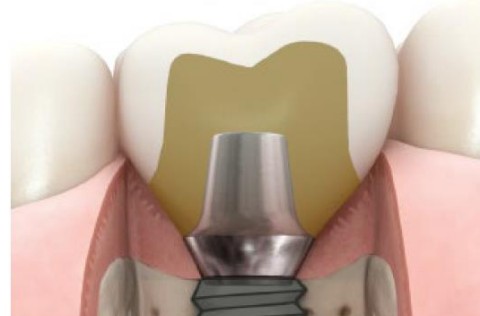

**Figure 1.** The main design differences between CAD/CAM custom abutment (**left**) and stock abutment (**right**). Courtesy of Dentsply IH [8].

**Table 1.** List of abbreviations used in the text.

| Abbreviation | Definition |
| --- | --- |
| CAD/CAM | computer-aided design/computer-aided manufacturing (milling) |
| PI | plaque index |
| mPI | modified plaque index |
| API | average plaque index |
| mAPI | modified average plaque index |
| i-API | initial average plaque index |
| 4m-API | four-month average plaque index |
| 8m-API | eight-month average plaque index |
| 12m-API | twelve-month average plaque index |
| mBI | modified bleeding index |
| ABI | average bleeding indeks |
| mABI | modified average bleeding index |
| i-ABI | inital average bleeding index |
| 4m-ABI | four-month average bleeding index |

**Table 1.** *Cont.*

| Abbreviation | Definition |
|---|---|
| 8m-ABI | eight-month average bleeding index |
| 12m-ABI | twelve-month average bleeding index |
| 8m-ABI-D | eight-month average bleeding index dichotomized |
| 12m-ABI-D | twelve-month average bleeding index dichotomized |
| PPD | pocket probing depth |
| APPD | average probing pocket depth |
| i-APPD | initial pocket probing depth |
| 4m-APPD | four-month pocket probing depth |
| 8m-APPD | eight-month pocket probing depth |
| 12m-APPD | twelve-month pocket probing depth |
| CBLE | crestal bone loss evaluation |
| CBLEm | crestal bone loss evaluation mesialy |
| CBLEd | crestal bone loss evaluation distaly |
| ACBLE | average crestal bone loss evaluation |
| TCBLE | total crestal bone loss evaluation |
| i-ACBLE | inital average crestal bone loss evaluation |
| 12m-ACBLE | twelve-month average crestal bone loss evaluation |
| SH | smoking habit |
| GB | gingival biotype |
| OH | oral hygiene |
| AM | arithmetic mean |
| SD | standard deviation |
| CI | confidence interval |

CAD/CAM technology consists of computer software reproducing the implant position and enabling abutment design with an ideal shape and inclination. This information is then electronically transmitted to the CAM milling machine, which fabricates the designed abutments from a block of the desired material [9,10]. A recent systematic review of the literature reported the same quality of CAD/CAM technology for the fabrication of dental implant abutments [11]. Custom abutments fabricated by means of CAD/CAM technology enable advantages of both stock and laboratory cast abutments by eliminating their disadvantages. CAD/CAM technology eliminates inevitable dimensional inaccuracies of conventional laboratory fabrication that result from the procedures of waxing-up, investing, casting, and polishing, as well as the influence on the implant-abutment connection and dependence on knowledge and skill of a dental technician [4]. CAD/CAM technology assures the homogeneity of titanium abutments with optimal material features [12]. In addition, unlike stock abutments, CAD/CAM custom abutments are not subject to subsequent changes, and their surface remains intact after fabrication. Dental technicians design abutments by means of CAD software with built-in control parameters. There are five main CAD/CAM systems for the fabrication of custom implant abutments on the market today (Procera, Nobel Biocare; Encode, Biomet 3i; Cares, Straumann; Etkon, Straumann; and Atlantis, Dentsply Implants) [13].

Implant abutments can be fabricated from titanium, precious metal alloys, and zirconium oxide ceramics [4]. Previous research has not found differences between titanium abutments and zirconia abutments regarding the reaction of soft peri-implant tissues [14]. Systemic analysis from the published research has shown that titanium is a highly reliable

material for abutment fabrication [15]. However, the greatest disadvantage of titanium is the shining of its dark grey colour through peri-implant soft tissue, which is aesthetically unacceptable [16]. On the other hand, zirconia abutments offer a much better aesthetic result, especially in the case of thin peri-implant mucosa [17], but the fragility of ceramics is one of its main disadvantages [14].

The peri-implant bone level is one of the most important criteria in the evaluation and monitoring of peri-implant tissue health [18]. Early cortical bone loss occurs when the implant is exposed in the oral cavity and results from the remodelling process associated with the establishment of the biologic width [19], whereas subsequent bone loss is mainly caused by bacterial colonisation and subsequent infiltration of inflammatory cells, which leads to the destruction of peri-implant tissues [18]. Radiological bone loss of 1.5 mm during the first year under loading, accompanied by bone loss of 0.2 mm per year, is a satisfactory criterion for determining implant success [18]. Several factors adversely affect the remodelling process and result in cortical bone resorption. They include: (1) traumatic surgical implant placement [20]; (2) excessive loading [21]; (3) microbiologic contamination of microleakage between the implant and implant abutment [22]; (4) micromotions of the implant and implant abutment [23]; and (5) repeated fastening and loosening of screws. The exclusion of one or more of these factors is desirable for implant–prosthetic treatment success [24].

The aim of this study was to compare clinical and radiographic parameters of monitoring by means of standard procedures and measurements that are applied in similar research between subjects with stock and custom CAD/CAM abutments during one year after functional loading of dental implants by prosthetic replacements.

## 2. Materials and Methods

This study was approved by the Ethical Committee, School of Dental Medicine, University of Zagreb, Croatia. The sample size was calculated for the before–after study (paired *t*-test) with the following parameters: $\alpha = 5.0\%$; $\beta = 20.0\%$; E = 0.500; and SD = 0.90, resulting in group size N = 28 ($\alpha$—Type I error rate; $\beta$—Type II error rate; E—Effect size; SD—Standard Deviation). A total of 64 subjects of both genders had a single tooth loss restored by a dental implant and prosthetic suprastructure. By means of a two-phase surgical technique, subjects were provided with dental implants ANKYLOS C/X® (DENTSPLY IH GmbH, Karlsruhe, Germany), which were, according to the surgical protocol, recommended by the manufacturer, placed 1 mm subcrestally [25]. The diameters of the placed implants were 3.5 mm and 4.5 mm, and lengths were 8 mm to 14 mm. Smoking habit, oral hygiene level, and gingival biotype were also recorded. Gingival biotype was assessed according to the transparency technique of marginal gingiva by means of a periodontal probe [26]. After 12 weeks of healing, healing abutments were placed. After four weeks of soft tissue healing, customised metal–ceramic crowns were fabricated. All subjects were randomly assigned to one of the research groups. In one group of subjects, a total of 34 stock abutments of the same manufacturer were used, and in the second group of subjects, 30 of them, custom CAD/CAM abutments fabricated by means of the ATLANTIS® (DENTSPLY IH GmbH, Mölndal, Sweden) system. All abutments were fabricated from a titanium alloy. Both abutment types were made of the titanium alloy Ti6Al-4V (Grade 5), which consists of 90% titanium, 6% aluminium, and 4% vanadium.

When designing custom abutments in the ATLANTIS® system, it is possible to determine in more detail the appearance and dimensions of the future abutment prior to its milling from a titanium alloy block. This way, abutments of a fully anatomic shape were selected for all subjects, with straight contours of the subgingival parts of abutments and a rounded shoulder on the gingival edges of abutments. The rounded shoulder position was determined in the way that it was placed on the buccal surface 1.7 mm and on the mesial, distal, and oral surface of the abutment 0.7 mm under the marginal edge of soft tissues.

The prosthetic restoration was fixed with Premier Implant Cement (Premier Products Co., Arnold, MO, USA). Subjects were instructed about maintaining proper oral hygiene

around dental implants. All teeth, as well as dental implant crowns, should be brushed at least twice a day using low-abrasive toothpaste and flossed at least once a day. Flossing should be performed with dental floss or a water flosser. The subjects were instructed that, if possible, they should brush their teeth after every meal, paying special attention to the sides of the implant and interdental spaces.

Examination of soft tissues around dental implants included determination of modified plaque index (mPI) and modified bleeding index (mBI) according to Mombelli et al. [27], as well as measurement of pocket probing depth (PPD) by means of a plastic periodontal probe with a 12 mm scale (UNC 12 ColorVue©, Hu-Friedy, Des Plaines, IL, USA). All three parameters were measured on the buccal, lingual, mesial, and distal side of an implant-supported metal–ceramic crown. Following intra-examiner reliability testing with an ICC (intraclass correlation coefficient) > 0.95, the same experienced clinician (I. P.) conducted all measurements. When determining mPI and mBI, each of the four mentioned surfaces was assessed with a mark from 0 to 3. As a criterion for assessment of the mPI, the following classification was used: 0—no plaque; 1—plaque detected only by moving the tip of the probe over the smooth crown surface along the gingival edge; 2—plaque visible to the naked eye; and 3—an abundance of soft deposits. When determining the mBI, the following classification was applied: 0—no bleeding; 1—individual spot bleeding visible; 2—bleeding forming a thin continual red line on the gingival edge; 3—abundant bleeding. The average modified plaque index (mAPI) was calculated from the measured values as the average measurement value on all four crown surfaces (e.g., 0 + 1 + 1 + 0/4; mAPI = 0.5). The average modified bleeding index (mABI) was calculated in the same way. The state of supporting tissues around the implant, i.e., the crown, was assessed by measuring the pocket probing depth (PPD). The depth measured by a periodontal probe is determined as the distance from the gingival edge to the tip of the probe, which entered the sulcus by a moderate probing force. Measurement was performed on the mesial, distal, vestibular, and lingual crown surfaces, and the average values of measured pocket probing depth (APPD) on all four measured points were calculated as their average value. The values mPI, mBI and PPD, i.e., mAPI, mABI and APPD, were measured during the first check-up (a week after cementing the crown) and at check-ups after four (4m-), eight (8m-), and twelve (12m-) months.

The radiologic assessment of the alveolar bone level around the dental implant (CBLE; crestal bone loss evaluation) was performed based on digital radiographic images made immediately after cementing the crown, and control recordings were made after twelve months. All radiographic recordings were made by the FOCUS® device and corresponding digital sensor SNAPSHOT® (Instrumentarium Dental, Helsinki, Finland) by applying the recording technique adopted by Brägger et al. [28]. Measuring was performed in the computer program CliniView® (Instrumentarium Dental, Helsinki, Finland). Crestal bone level evaluation on the mesial (CBLEm) and distal side of the dental implant (CBLEd) was measured, and the implant shoulder (platform) was taken as the reference point. Taking into account well-researched and documented properties of the ANKYLOS® implant system with respect to the stability of the cortical bone around the implant [29], on radiographic images, the bone level that was situated more apically than the implant shoulder achieved in measurements positive values 'bone loss'), expressed in millimetres, and in cases where the implant shoulder was covered by a layer of bone tissue, measurements achieved negative values ('bone gain'). Since the length and the diameter of the placed dental implant are known, measurements based on digital radiographic images served first for calibration of measurements in relation to the known length and diameter of the placed implant. Afterwards, a reference line was drawn through the implant shoulder, which is perpendicular to the longitudinal axis of the implant, and the distance of the bone level to the reference line was measured (Figure 2).

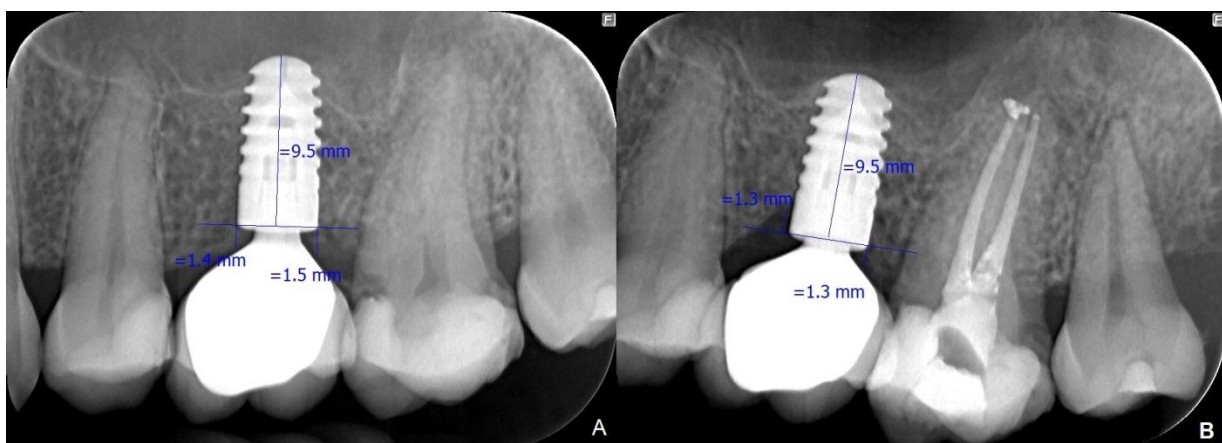

**Figure 2.** The principle of alveolar bone level measurement on digital radiographic images. Differences between the initial measurement (**A**) and control measurement after 12 months (**B**) can be observed.

The average crestal bone loss evaluation (ACBLE) was calculated as the average value of measurement from the mesial and distal sides of the implant at the initial measurement (i-ACBLE) and at the control measurement after twelve months (12m-ACBLE). The value of total crestal bone loss evaluation change (TCBLE) was also calculated as the difference of values of average crestal bone loss evaluation at measurement after twelve months and at initial measurement (TCBLE = 12m-ACBLE–i-ACBLE). The positive variable values TCBLE during the observed period of twelve months indicated a new bone development and negative values pointed to bone resorption around dental implants.

Statistical analysis was performed by use of STATISTICA 12 (StatSoft. Inc., Tulsa, OK, USA), and levels lower than 0.05 were considered significant. When needed, data were analysed by the $\chi^2$ test or by Fisher's exact test. The normality of the distribution of quantitative measurements (properties) was tested by the Shapiro–Wilk test. Differences in the distribution of quantitative variables frequencies were analysed by the Student's *t*-test. Student's *t*-test for repeated measurements, Mann–Whitney's U test, and ANOVA test for repeated measurements and Friedmann's two-way analysis of variance for repeated measurements. The predictive values of the chosen variables on the ABI index were assessed by the logistic regression model (Enter method).

### 3. Results

Sixty-four subjects of both genders participated in this study: 31 male subjects at the age of 26 to 81 (mean age 44) and 33 female subjects at the age of 28 to 73 (mean age 40). The average age of all subjects, regardless of gender, was 44.4 ± 12.1 years (Table 2).

**Table 2.** Distribution of subjects according to type of abutment, gingival biotype (GB), smoking habits (SH), and oral hygiene level (OH).

| Type of Abutment | Number of Subjects | Gingival Biotype (GB) | | Smoking Habits (SH) | | Oral Hygiene Level (OH) | | |
|---|---|---|---|---|---|---|---|---|
| | | Thin | Thick | Yes | No | Poor | Good | Excellent |
| STOCK | 34 | 5 | 29 | 9 | 25 | 0 | 13 | 21 |
| CUSTOM | 30 | 6 | 24 | 11 | 19 | 0 | 12 | 18 |
| TOTAL | 64 | 11 | 53 | 20 | 44 | 0 | 25 | 39 |

Modified average plaque index values (mAPI), initial and after four, eight, and twelve months, have not indicated significant differences between the two groups of subjects. However, Friedmann's two-way analysis of variance for repeated measures showed that average values of all subjects for modified average plaque—initial (i-mAPI), after four

months (4m-mAPI), after eight months (8m-mAPI), and after twelve months (12m-mAPI) significantly differed ($\chi^2$ = 16.76 (3); $p$ = 0.001) (Table 3).

**Table 3.** Descriptive statistics and distribution of differences according to groups of subjects for variables i-mAPI (initial modified average plaque index), 4m-mAPI (4-month modified average plaque index), 8m-mAPI (8-month modified average plaque index), and 12m-mAPI (12-month modified average plaque index).

|  |  | **i-mAPI** | **4m-mAPI** | **8m-mAPI** | **12m-mAPI** |
|---|---|---|---|---|---|
| STOCK | Median | 0.25 | 0.125 | 0.25 | 0.5 |
|  | Range | 0–1.75 | 0–1.5 | 0–1.5 | 0–1.5 |
| CUSTOM | Median | 0 | 0.125 | 0 | 0.125 |
|  | Range | 0–1.5 | 0–1.5 | 0–1.25 | 0–1.5 |
| Mann–Whitney's U test | $p$ | 0.333 | 0.744 | 0.150 | 0.108 |
| Total for all subjects (STOCK + CUSTOM) | Median | 0.25 | 0.125 | 0 | 0.25 |
|  | Range | 0–1.75 | 0–1.5 | 0–1.5 | 0–1.5 |
| Friedman's two-way analysis of variance for repeated measures | $\chi^2$ (number of degrees of freedom) $p$ | $\chi^2$ = 16.76 (3); $p$ = 0.001 | | | |

Average modified bleeding index values—initial (i-mABI) and after four months (4m-mABI) revealed no significant differences between the two tested groups. However, there was a significant difference in the average bleeding index values after eight months (8m-mABI) between the two tested groups (Mann–Whitney's U test $p$ = 0.002). A significant difference (Mann–Whitney's U test $p$ < 0.001) was also observed for the average bleeding index values after twelve months (12m-mABI) between the two groups of subjects (Table 4).

**Table 4.** Descriptive statistics and distribution of differences according to groups of subjects for variables i-mABI (initial modified average bleeding index, 4m-mABI (4-month modified average bleeding index), 8m-mABI (8-month modified average bleeding index), and 12m-mABI (12-month modified average bleeding index).

|  |  | **i-mABI** | **4m-mABI** | **8m-mABI** | **12m-mABI** |
|---|---|---|---|---|---|
| STOCK | Median | 0.25 | 0.125 | 0.5 | 0.5 |
|  | Range | 0–2 | 0–2.5 | 0–1.25 | 0–1.75 |
| CUSTOM | Median | 0.25 | 0 | 0 | 0 |
|  | Range | 0–1.25 | 0–0.75 | 0–0.5 | 0–0.5 |
| Mann–Whitney's U test | $p$ | 0.427 | 0.132 | 0.002 | <0.001 |
| Total for all subjects (STOCK + CUSTOM) | Median | 0.25 | 0 | 0.25 | 0.25 |
|  | Range | 0–2 | 0–2.5 | 0–1.25 | 0–1.75 |
| Friedman's two-way analysis of variance for repeated measurements | $\chi^2$ (number of degrees of freedom) $p$ | $\chi^2$ = 3.15 (3); $p$ = 0.369 | | | |

There was no significant difference in the mean values of pocket probing depths between the two groups of subjects (Table 5).

**Table 5.** Descriptive statistics and distribution of differences according to groups of subjects for variables i-APPD (initial average pocket probing depth), 4m-APPD (4-month average pocket probing depth), 8m-APPD (8-month average pocket probing depth) and 12m-APPD (12-month average pocket probing depth).

|  |  | i-APPD | 4m-APPD | 8m-APPD | 12m-APPD |
|---|---|---|---|---|---|
| STOCK | AM | 2.53 | 2.67 | 2.62 | 2.6 |
|  | SD | 0.83 | 0.81 | 0.76 | 0.76 |
| CUSTOM | AM | 2.55 | 2.58 | 2.64 | 2.66 |
|  | SD | 0.58 | 0.64 | 0.59 | 0.56 |
| Student's *t*-test for independent samples | T | −0.09 | 0.44 | −0.16 | −0.38 |
|  | P | 0.928 | 0.657 | 0.870 | 0.709 |
| Total for all subjects | AM | 2.53 | 2.62 | 2.63 | 2.62 |
|  | SD | 0.72 | 0.73 | 0.68 | 0.66 |
| ANOVA for repeated measurements with Greenhouse–Geisser correction | F (number of degrees of freedom) *p* | F = 2.17 (1.45; 91.24) *p* = 0.135 | | | |

Total bone level reduction around the implant (−0.91 CI 95% −0.14–−0.04) in all subjects between i-ACBLE (initial average crestal bone loss evaluation) and 12m-ACBLE (12-month average crestal bone loss evaluation) was significant ($t = -3.7$; $p < 0.001$) (Table 6).

**Table 6.** Descriptive statistics and distribution of differences according to groups of subjects for variables i-ACBLE (initial average crestal bone loss evaluation) and 12m-ACBLE (12-month average crestal bone loss evaluation).

|  |  | i-ACBLE | 12m-ACBLE |
|---|---|---|---|
| STOCK | AM | −0.32 | −0.24 |
|  | SD | 0.88 | 0.94 |
| CUSTOM | AM | −0.27 | −0.15 |
|  | SD | 0.47 | 0.54 |
| Student's *t*-test for independent samples | T | −0.29 | −0.45 |
|  | *p* | 0.775 | 0.652 |
| Total for all subjects | AM | −0.29 | −0.2 |
|  | SD | 0.71 | 0.78 |
| Student's *t*-test for dependent samples | *t* | −3.7 | |
|  | *p* | <0.001 | |

Significant differences between study groups for variables 8m-mABI and 12m-mABI were found (see Table 4), predictive values for stock and custom abutments (Table 7), gingival biotype, and oral hygiene level were examined for variables 8m-mABI and 12m-mABI by means of logistic regression model and Enter method (Table 8). It was necessary to dichotomise the variable 8m-mABI, i.e., 12m-mABI, in the way that the subjects were grouped into two groups, depending on the occurrence of sulcus probing bleeding. New variables were denoted as 8m-mABI-D and 12m-mABI-D, where the suffix -D stands for dichotomised, and their values were 0 for cases without bleeding and 1 for cases with bleeding (Table 9).

**Table 7.** Descriptive statistics and distribution of differences according to groups for variable TCBLE (total crestal bone loss evaluation).

| | | | TCBLE |
|---|---|---|---|
| STOCK | | AM | 0.05 |
| | | SD | 0.17 |
| CUSTOM | | AM | 0.11 |
| | | SD | 0.15 |
| Student's *t*-test for independent samples | | T | 1.66 |
| | | *p* | 0.105 |
| Total for all subjects (N = 63) * | | Average | 0.76 |
| | | SD | 0.16 |

* Values of variable TCBLE (total crestal bone loss evaluation) in one subject from the group with stock abutment had extreme deviation and affected the normality of distribution of values so that the subject was excluded from statistical processing for variable TCBLE.

**Table 8.** Descriptive statistics and distribution of differences for variable TCBLE (total crestal bone loss evaluation) in relation to variables smoking habit (SH), gingival biotype (GB), and oral health level (OH).

| Variable | Value | N | TCBLE AM | TCBLE SD | Student's *t*-Test |
|---|---|---|---|---|---|
| SH | YES | 20 | 0.01 | 0.14 | *t* = 1.94 |
| | NO | 43 | 0.11 | 0.13 | *p* = 0.028 |
| GB | Thin | 11 | 0.13 | 0.24 | *t* = 1.25 |
| | Thick | 52 | 0.06 | 0.14 | *p* = 0.218 |
| OH | Good | 24 | 0.09 | 0.13 | *t* = 0.58 |
| | Excellent | 39 | 0.07 | 0.17 | *p* = 0.564 |

**Table 9.** Differences in 8m-mABI-D (8-month modified average bleeding index, dichotomised and 12m-mABI-D D (12-month modified average bleeding index, dichotomised) regarding the type of abutment, oral hygiene level, and gingival biotype.

| | | 8m-mABI-D | | 12m-mABI-D | |
|---|---|---|---|---|---|
| | | 0 | 1 | 0 | 1 |
| Type of abutment | STOCK | 11 | 23 | 6 | 28 |
| | CUSTOM | 18 | 12 | 17 | 13 |
| | Total | 29 | 35 | 23 | 41 |
| | Pearson χ² | 4.92 | | 10.54 | |
| | *p* | 0.027 | | 0.001 | |
| Oral hygiene | Good | 7 | 18 | 5 | 20 |
| | Excellent | 22 | 17 | 18 | 39 |
| | Total | 29 | 35 | 23 | 41 |
| | Pearson χ² | 4.96 | | 4.53 | |
| | *p* | 0.026 | | 0.033 | |
| Gingival biotype | Thin | 6 | 5 | 3 | 8 |
| | Thick | 23 | 30 | 20 | 33 |
| | Total | 29 | 35 | 23 | 41 |
| | Fischer's exact test | 0.526 | | 0.732 | |
| | *p* | | | | |

The influence of gingival biotype was not significant for the examined variables ($p = 0.526$; $p = 0.732$). The influence of the abutment type on variables 8m-mABI-D (8-month modified average bleeding index, dichotomised) and 12m-mABI-D (8-month modified average bleeding index, dichotomised) was significant ($p = 0.027$; $p = 0.001$), as well as the influence of oral hygiene ($p = 0.026$; $p = 0.033$) (Table 9).

Logistic regression model was confirmed as significant: $\chi^2 = 10.733$ (2); $p = 0.005$ in case of 8m-mABI-D, i.e., $\chi^2 = 16.889$ (2); $p = 0.000$ in case of 12m-mABI-D. According to the Wald $\chi^2$ test of coefficient significance, both predictors (type of abutment and oral hygiene level) were significant for variables 8m-mABI-D (Nagelkerke $R^2 = 0.206$) and 12m-mABI-D (Nagelkerke $R^2 = 0.318$) (Table 10).

**Table 10.** Results of logistic regression model (Enter method) with a review of predictive values of variables type of abutment and oral hygiene level in relation to variables 8m-mABI-D and 12m-mABI-D (8-month and 12-month modified average bleeding index dichotomised).

| | | Coefficient | Statistical Error | Wald (df) | *p* | Chance Ratio | 95% CI |
|---|---|---|---|---|---|---|---|
| **8m-mABI-D** | Type of abutment | 1.29 | 0.56 | 5.29 (1) | 0.022 | 3.62 | 1.2–10.82 |
| | Oral hygiene | 1.35 | 0.59 | 5.28 (1) | 0.022 | 3.85 | 1.22–12.15 |
| | Constant | −0.98 | 0.47 | 4.3 (1) | 0.038 | 0.38 | |
| **12m-mABI-D** | Type of abutment | 2.04 | 0.63 | 10.36 (1) | 0.001 | 7.65 | 2.22–26.41 |
| | Oral hygiene | 1.55 | 0.67 | 5.31 (1) | 0.021 | 4.7 | 1.26–17.51 |
| | Constant | −0.91 | 0.49 | 3.5 (1) | 0.061 | 0.4 | |

## 4. Discussion

There are not many published studies regarding the comparison between custom CAD/CAM dental implant abutments and stock abutments, especially with respect to their influence on peri-implant tissues. Although the advantages of custom CAD/CAM abutments are well-known, there are only a few data regarding their influence on peri-implant tissues.

Apicella et al. [30] compared the custom CAD/CAM and stock abutments and titanium and zirconia as abutment materials based on radiographic images and scanning electronic microscope images regarding fit at the implant-abutment interface. Fit at the implant-abutment interface is critical for the long-term success of implant–prosthetic treatment [12]. A reliable and precise fit is desirable in order to increase the maximum mechanical stability of the suprastructure [31] and in order to avoid possible associated biological complications [32]. Namely, it has been noted that leakage at the implant-abutment interface can increase the amount of stress on prosthetic components, implant, and peri-implant bone [33]. The abutment margin can also accelerate bacterial accumulation and, in this way, be a source of inflammation of the surrounding soft tissues [34]. Furthermore, it is known that bacterial leakage at the implant-abutment interface can have an etiologic role in peri-implantitis [35]. Since implant-abutment fit accuracy can influence the occurrence of biological and mechanical complications; it is of extreme importance for leakage at the junction between two prosthetic parts to be as small as possible and fit as precisely as possible. Apicella et al. [30] concluded that between custom CAD/CAM and stock abutments, there was no significant difference with respect to fit accuracy (two-year follow-up study). The same authors also reported that there was no significant difference in comparison of titanium and zirconia, custom CAD/CAM and stock abutments.

Hamilton et al. [1] also examined the fit of custom CAD/CAM and stock abutments on different implant systems. The same authors reported that the design of the abutment surface and its fabrication significantly influence the fit between the abutment and implant.

Lops et al. [36] concluded that custom CAD/CAM abutments were associated with better stability of peri-implant soft tissues. A significant difference was found when titanium CAD/CAM abutments were compared with titanium stock abutments.

Little data regarding middle-term monitoring of the gingival level around implants is available at the moment. Gingival biotype and implant shoulder level are supposed to relate to soft tissue recession [37]. On the contrary, a literature review by Cairo et al. [38] showed that these parameters were not connected with soft tissue recession around implants placed in the anterior region. Unfortunately, there are no available data regarding the comparison between soft tissue stability around implants with custom CAD/CAM and stock abutments in the anterior region.

Experimental research and research on animals confirmed that the formation and development of microbial biofilm represents an important etiologic factor in the pathogenesis of peri-implant diseases. The level of oral hygiene in subjects within this research was satisfactory. The plaque index did not indicate a significant difference between the examined groups. Friedman's two-way analysis of variance for repeated measurements indicated that the average values of all subjects for i-API, 4m-API, 8m-API, and 12m-API were significantly different ($\chi^2 = 16, 76 (3)$; $p = 0,001$). The initial values of API measured a week after suprastructure cementation might be a result of tissue irritation during cementation and consequent sensitivity of this area to brushing.

At the follow-up after four months, a decrease in API values was observed, and they achieved 0 at the follow-up after eight months, which is probably a result of the patient's motivation and given detailed instructions about oral hygiene. On the other hand, follow-up after twelve months showed an increase in API and the need for continual follow-ups that would motivate patients. Within this research, an examination of the influence of oral hygiene on crestal bone loss was performed, but no significant correlation was established. It seems that subjects should be monitored for a longer period. On the other hand, a significant correlation between oral hygiene and average bleeding on probing after eight ($p = 0.026$) and twelve months ($p = 0.033$) was noticed. The results of the logistic regression model indicated poor oral hygiene as a predictor for the occurrence of bleeding on probing.

Increased accumulation of dental plaque leads to a stronger inflammatory reaction in peri-implant soft tissues, which can be objectively assessed by means of bleeding on the probing index. This parameter has a central role in monitoring changes in peri-implant tissue health. It is important to mention that inflammation does not necessarily imply an infection [39].

The results of this study showed a significant difference in bleeding on probing between two groups of subjects, which increased with time. Namely, the results of Mann–Whitney's U test showed significant differences in the values 8m-ABI and 12m-ABI ($p < 0.001$) between the two groups of subjects ($p = 0.02$). The logistic regression model also showed that the abutment type has a significant impact on the ABI-D after eight ($p = 0.022$) and after twelve months ($p = 0.001$). Obviously, custom abutments fit better into the biological environment and irritate peri-implant soft tissues less, thus reducing inflammation. Based on these results, it can also be concluded that soft tissues needed eight months for adaptation to the newly developed conditions after placement of the restoration.

Differences in the soft tissue composition, organisation, and attachment between the gingiva and root surface, on the one hand, and peri-implant mucosa and implant surface, on the other hand, make a direct comparison of probing depth measurements around teeth and around implants more difficult [40]. The form and surface of implants also influence the periodontal probe penetration. Probing peri-implant tissues around some implants is impossible due to their design characteristics (concavities, suprastructure shoulder). Lack of a smooth surface, as is the case with implants with a cover-coating of titanium plasma, sandblasted or acid-etched, can disturb periodontal probe penetration, which can cause underestimation of probing depth values [41]. The values of periodontal pocket depth must be interpreted also in the context of surgical positioning of implants. A progressive increase in probing depth is an alarming sign. In accordance with this, measurement of the

initial values of probing depth at the time of prosthetic suprastructure delivery is of critical importance because it enables comparison with future measurements [42]. If high values of probing depth relate to peri-implant soft tissue recession, then probing depth would not exactly reflect the cortical bone loss around the implant. Brägger et al. [28] reported that the level of connective attachment, in combination with radiographic indicators collected after two years of functional loading of implants, is a good predictor of the peri-implant tissue state. Several studies reported that pocket-depth measurement around implants is a good indicator of crestal bone loss [43] and that a progressive increase in probing depth is an alarming sign. In this study, the measurement of sulcus-probing depth (APPD) did not indicate a significant difference between subjects with stock and custom abutments. Generally, all measurements indicated values less than 3 mm, and previous research data reported that successful implant–prosthodontic rehabilitation allows a sulcus-probing depth of about 3 mm [27].

Long-term preservation of crestal bone height around osseointegrated implants is often used as a primary criterion of different implant systems' success. Crestal bone loss of less or equal to 1.5 mm during the first year of functional loading and less or equal to 0.2 mm per year after the first year of functional loading was proposed as one of the criteria for the successful implant–prosthodontic treatment [44]. This criterion was questioned as longitudinal studies gave proof that crestal bone loss around osseointegrated implants in patients during follow-up can be minimal [45]. Conventional radiography is a widely accepted technique for long-term evaluation of approximal height changes of cortical bone. The long-cone paralleling technique is generally used with the help of a positioning holder. It should be noted that usual radiography gives a high percentage of false negative results, i.e., it has a low sensitivity for early detection of bone remodelation and pathologic changes [46].

Therefore, radiological procedures should be considered after clinical parameter assessment [47]. The distance from the implant shoulder, in relation to the alveolar bone crest, represents a reliable parameter for long-term monitoring in clinical practice [42]. It should also be noted that radiographic proof of bone contact with an implant does not necessarily imply osseointegration at the histologic level [48]. Computer-aided analysis of the image showed higher diagnostic accuracy (higher sensitivity) for the detection of minimal changes in periodontal tissue [46]. The results of this study for all subjects in total established significant initial loss of total bone level (difference between i-ACBLE and 12m-ACBLE) ($p < 0.001$), which is in accordance with the results of Zembic et al. [49] and Chang et al. [50], who indicated that changes in soft and hard peri-implant tissues occur mainly during the first six months. No significant difference between the two examined groups, i.e., custom CAD/CAM and stock abutments regarding cortical bone loss, was found. Lin et al. [51] found remarkably similar outcomes in their research. Using digital periapical radiographs, this study attempted to determine the difference between CAD/CAM custom abutments and original stock abutments based on the change in mesial and distal bone levels. The vertical marginal bone levels of the mesial and distal surrounding implant bones were measured using radiographs taken prior to delivery, following functional loading for one month, and at 3, 6, and 12 months. For a total of 57 implants in 50 patients, 22 CAD/CAM custom abutments and 35 original stock abutments were utilised. There was no significant difference in the bone levels of custom abutments and stock abutments over any period.

Seldom have clinical investigations compared CAD/CAM custom titanium abutments to original 1-piece prefabricated titanium stock abutments for posterior fixed dental prostheses. Hsiao et al. carried out a retrospective research study with roughly seven years of follow-up. In the evaluation of 99 patients with 195 implants, implant failure was not observed in either group. There were no significant differences in the incidence of ceramic chipping, peri-implantitis, peri-implant mucositis, or mean marginal bone loss between the two groups. Using the CAD/CAM or stock abutments to support a posterior fixed dental prosthesis on dental implants did not affect the occurrence of biological complications.

CAD/CAM abutments demonstrated a greater abutment screw loosening rate and a lower decementation rate than stock abutments [52].

Another retrospective study assessed hard and soft tissue reactions as well as mechanical and technical problems around CAD/CAM abutments made of titanium, gold-hue titanium, and zirconia. This research included 123 patients with 291 CAD/CAM abutments who were monitored for at least two years. Each year, clinical and radiographic data were evaluated, and complications were documented. There were no reported implant or restoration failures. There was one zirconia abutment fracture. The four-year survival rate for restorations and abutments was 100 per cent and 99.66 per cent, respectively. There were no significant differences between the biological and radiographic indices. The bleeding on the probing index was positive at 42% of implant sites, and it had no significant correlation with the overall change in marginal bone level (0.02 mm) of bone gain [53].

It is well known that smoking affects both the osseointegration process, maintenance of the crestal bone level and consequently with cortical bone loss [54]. The results of this study showed a significant adverse effect of smoking on the total assessment of crestal bone loss ($p = 0.028$). In a recent retrospective cohort study involving 4591 dental implants, statistical analysis revealed that there was initially no difference in crestal bone loss between heavy smokers and non-smokers. However, a significant main effect ($p < 0.01$) and a significant interaction with time were discovered. After 4 years of function, marginal bone loss among heavy smokers was more rapid [55]. This stress the importance of patient selection for implant–prosthodontic treatment and of long-year postoperative monitoring.

Different gingival biotypes react differently to inflammation, trauma, parafunctional activities, as well as prosthodontic replacements [56]. Ferrari et al. [57], who examined the influence of abutment materials on dental implants, including titanium alloys, the so-called 'gold-hue titanium' and zirconia, reported that abutment material in the period of two years did not affect peri-implant soft tissues, such as buccal recession depth, sulcus probing depth, depth of keratinised gingiva, and radiographic bone height mesially and distally. The initial values of the examined variables associated with soft tissues could not predict the possible occurrence of recessions after two years. We also found no significant correlation of gingival biotype with either bleeding on probing or total assessment of crestal bone loss.

In this study, the limitations included the prospective study design, the size of the sample population, the use of two-dimensional digital radiographs, the one-year follow-up period after functional loading, and the absence of intraoral occlusion data to confirm the existence of overloading, which was reported as a potential risk factor for implant failure. The use of two-dimensional radiography was restricted to measuring mesiodistal marginal bone loss. Hence, it was unable to quantify the buccolingual bone level and crestal bone volume surrounding the dental implants. Using cone-beam computed tomography [58], the change in buccolingual marginal bone loss and bone volume around implants could be evaluated more objectively. The data integrity could be strengthened by increasing the sample size and extending the follow-up period, as well as conducting more future clinical trials on this topic.

## 5. Conclusions

Within the limitations of this study, there was a significant difference between subjects with stock and custom CAD/CAM abutments regarding the average modified bleeding index after eight and twelve months upon cementing a prosthetic restoration. The average modified bleeding index was lower for subjects with custom CAD/CAM abutments than for subjects with stock abutments. The influence of abutment type and oral hygiene level of bleeding on probing was significant after eight and twelve months of follow-up, with lower values of bleeding on probing for CAD/CAM abutments. The type of abutment and oral hygiene level were significant predictors of bleeding on probing after eight months and a 12-month follow-up period. There were no significant differences regarding average pocket probing depth between subjects with stock and custom CAD/CAM abutments.

However, the average pocket-probing depth in subjects with stock abutments increased in the first four months and decreased afterwards. As opposed to this, in subjects with custom CAD/CAM abutments, slightly increasing pocket probing depth values during the whole period of one year were noticed. The differences in average values of initial and 12-month values of alveolar crestal bone loss evaluation between the two examined groups were not significant. However, negative values of the mentioned variables were found in both groups of subjects, which points to the fact that there is bone growth on the implant shoulder. When observing all subjects independent of the type of abutment, the values of the initial average crestal bone loss evaluation and the 12-month average crestal bone loss evaluation significantly differed and decreased with respect to the amount of bone tissue on implant shoulders. The change in total crestal bone loss evaluation between initial and control measurements after twelve months revealed no significant differences. Total bone gain after one year in subjects with custom abutments was slightly higher when compared to the subjects with stock abutments but did not differ significantly. There was a significant difference between smoking habit, gingival biotype, and oral hygiene level regarding the total crestal bone loss evaluation variable values. Smoking habit influenced the total crestal bone loss evaluation variable significantly.

**Author Contributions:** Conceptualization, I.P. and I.Š.; methodology, N.D.; software, M.V.; validation, I.P., I.Š. and D.G.; formal analysis, M.V.; investigation, I.P.; resources, N.D.; data curation, I.Š.; writing—original draft preparation, I.P.; writing—review and editing, I.Š.; visualisation, M.V.; supervision, N.D.; project administration, D.G.; funding acquisition, I.P. All authors have read and agreed to the published version of the manuscript.

**Funding:** This research received no external funding.

**Institutional Review Board Statement:** This study was conducted according to the guidelines of the Declaration of Helsinki and approved by the Institutional Review Board (or Ethics Committee) of the School of Dental Medicine, University of Zagreb, Protocol code: 05-PA-26-9/2022.

**Informed Consent Statement:** Informed consent was obtained from all subjects involved in the study.

**Data Availability Statement:** The data presented in this study are available on request from the corresponding author.

**Conflicts of Interest:** The authors declare no conflict of interest.

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
