# Peer review of "Radiological and Periodontal Evaluation of Stock and Custom CAD/CAM Implant Abutments—A One-Year Follow-Up Study"

_prosthesis, doi:10.3390/prosthesis5020030_

Round 1
Author Response
Dear reviewer,
We'd like to thank you for all of your helpful suggestions, which we've used to make this new version of our manuscript. All corrections were made in Microsoft Word using the "Track Changes" option, so they are easily visible.
In general, a figure showing the main differences in design between stock and custom CAD/CAM abutments and a table listing all the abbreviations used in this manuscript were added.
A few new paragraphs were added to the Discussion part, and the References list was updated with more current research. The limitations of this study were also addressed at the end of the Discussion part.
The Conclusion part is completely rewritten to better represent the findings of this research.
Spelling and grammar checks were done by a native English speaker, with correction of some typographical errors (i.e., a p value of 0.005 instead of 0.05).
Still, there aren't many studies that compare the clinical performance of stock and custom CAD/CAM abutments. We hope that this study will shed some new light on this area of implant dentistry and give us some new ideas.
Regards
Reviewer 2 Report
This research study provides a comparison in radiological and periodontal tissue assessment between stock abutments and custom abutments (CAD/CAM )in dental implants. The researchers have studied 64 patients who received two types of implant abutments- stock and custom. After evaluating their clinical and radiological characteristics at the end of four, eight and twelve months, it was determined that bleeding on probing differed significantly at eight and twelve months in comparison between these two types, while crestal bone loss was insignificant. In the abstract, please add a statement showing your overall result as per the findings in a simple but clear statement. It will make the abstract more self-explanatory.
The strength of the study is that it is very well presented and statistics are described in detail. However, as the study takes into consideration only a small group of patients/subjects, it is important to conduct a similar study with a larger cohort of patients. Another factor which may impact the results are implant manufacturers and the type of implants/ abutments used. For generalizability, it is important that similar studies are conducted in other countries to correlate the findings.
Please make the following corrections-
Line 13: "moment"- This sentence can be re-phrased. Implant abutment selection is an important step in implant treatment to restore one or more missing teeth.
Line 14: CAD/CAM full form
Line 31: "moment"- please rephrase
Line 35,36: Rephrase the sentence- It can be written as stock abuts have ease of handling and they are relatively less expensive.
Line 40,41: This sentence can be rephrased as "many researchers" instead of many authors.
Line 43: This sentence can be written " custom abutments which are fabricated (instead of formed)
Line 44: Such an abutment provides (instead of acts) adequate support...
Line 48: CAD/CAM in full form here too, as it first appears in the main text here.
Line 61,62,63: Rephrase this line- submitted word here can be changed or line rephrased
Line 270: "Enter Method"- I assume you would like to enter the name of the method used to conduct these statistics.
Line 307: According to (35)- Either you can write a reference number only.
line 315: PI -full form
Author Response
Dear reviewer,
We'd like to thank you for all of your helpful suggestions, which we've used to make this new version of our manuscript. All corrections were made in Microsoft Word using the "Track Changes" option, so they are easily visible.
Regards
Author Response
Dear reviewer,
We'd like to thank you for all of your helpful suggestions, which we've used to make this new version of our manuscript. All corrections were made in Microsoft Word using the"Track Changes" option, so they are easily visible.
The Conclusion part is completely rewritten to better represent the findings of this research.
Regards
Reviewer 4 Report
The authors compare clinical and radiographic parameters of between subjects with stock and custom CAD/CAM abutments during one year after functional loading of dental implants by prosthetic replacements. While the manuscript is generally well executed, there are several issues that should be addressed before further consideration for publication.
1. Were the stock and custom abutments made of same material? What is the composition of the titanium alloy? This should be detailed in the manuscript.
2. What is the characteristics of the abutments, any characterisation done? The properties, such as surface roughness, should have effect on the performance and should be discussed.
3. How detailed are the custom abutments? How they differ from the stock abuments?
4. Any consideration for other manufacturing techniques, such as additive manufacturing for production of custom abutments?
Author Response
Dear reviewer,
We'd like to thank you for all of your helpful suggestions, which we've used to make this new version of our manuscript. All corrections were made in Microsoft Word using the "Track Changes" option, so they are easily visible.
In general, a figure showing the main differences in design between stock and custom CAD/CAM abutments and a table listing all the abbreviations used in this manuscript were added.
Still, there aren't many studies that compare the clinical performance of stock and custom CAD/CAM abutments. We hope that this study will shed some new light on this area of implant dentistry and give us some new ideas.
Regards
Reviewer 5 Report
Dear Authors the article is interesting and fits the objectives of the journal
-Please be sure to use only keywords accordingly to medical subject headings (Mesh word) for a better indexing.
- The introduction section is very short and is needed to add other references to increase the quality of the manuscript, Add recent references about the topic of the article, dwelling in the introduction on articles published in 2022 and describing what your article will add compared to the last articles published; Preferably a published articles should be with 90 or more references.
I suggest you some articles
Telescopic overdenture on natural teeth: Prosthetic rehabilitation on OFD syndromic patient and a review on available literature PubMed ID 29460531
Teledentistry in the Management of Patients with Dental and Temporomandibular Disorders Doi: https://doi.org/10.1155/2022/7091153
Prosthodontic Treatment in Patients with Temporomandibular Disorders and Orofacial Pain and/or Bruxism: A Review of the Literature https://doi.org/ 10.3390/prosthesis4020025
-You need to review the grammar and English of your article, with the help of a native English speaker (you can specify who helped you in reviewing English in the acknowledgements) or simply by using a site that can support you in English
-I suggest you to add an image in order to improve the iconography of the article.
-I suggest you add a table with the list of abbreviations used in the text.
-Please expand conclusion section with main results and future perspectives of this study
Thank You,
Kind Regards
Author Response
Dear reviewer,
We'd like to thank you for all of your helpful suggestions, which we've used to make this new version of our manuscript. All corrections were made in Microsoft Word using the "Track Changes" option, so they are easily visible.
In general, a figure showing the main differences in design between stock and customCAD/CAM abutments and a table listing all the abbreviations used in this manuscript were added.
A few new paragraphs were added to the Discussion part, and the References list was updated with more current research.
The limitations of this study were also addressed at the end of the Discussion part.
The Conclusion part is completely rewritten to better represent the findings of this research.
Spelling and grammar checks were done by a native English speaker, with correction of some typographical errors (i.e., a p value of 0.005 instead
of 0.05).
The suggestion to include some published articles in this manuscript we could not fulfil because they are completely out of the scope of this study (e.g.,
temporomandibular disorder, bruxism, and overdentures).
Still, there aren't many studies that compare the clinical performance of stock and custom CAD/CAM abutments. We hope that this study will shed some new light on this area of implant dentistry and give us some new ideas.
Regards
Round 2
Reviewer 4 Report
NA
Author Response
Thank you for your valuable comments.